# A latent class assessment of healthcare access factors and disparities in breast cancer care timeliness

Matthew R. Dunn [1,2]*, Didong Li [3], Marc A. Emerson[1,2], Caroline A. Thompson[1,2], Hazel B. Nichols [1,2], Sarah C. Van Alsten [1,2], Mya L. Roberson [2,4], Stephanie B. Wheeler[2,4], Lisa A. Carey[2], Terry Hyslop[5], Jennifer Elston Lafata [2,6‡], Melissa A. Troester[1,2,7‡]

1 Department of Epidemiology, University of North Carolina, Chapel Hill, North Carolina, United States of America, 2 Lineberger Comprehensive Cancer Center, University of North Carolina, Chapel Hill, North Carolina, United States of America, 3 Department of Biostatistics, University of North Carolina, Chapel Hill, North Carolina, United States of America, 4 Department of Health Policy and Management, University of North Carolina, Chapel Hill, North Carolina, United States of America, 5 Thomas Jefferson University, Sidney Kimmel Cancer Center, Philadelphia, Pennsylvania, United States of America, 6 Eshelman School of Pharmacy, University of North Carolina, Chapel Hill, North Carolina, United States of America, 7 Department of Pathology and Laboratory Medicine, University of North Carolina, Chapel Hill, North Carolina, United States of America

‡ These authors are co-senior authors on this work.
* mdunn@unc.edu

**Data Availability Statement:** Code for the statistical analysis of this study can be found at https://github.com/mattdunnepi/bcdelays, and the

## Abstract

### Background

Delays in breast cancer diagnosis and treatment lead to worse survival and quality of life. Racial disparities in care timeliness have been reported, but few studies have examined access at multiple points along the care continuum (diagnosis, treatment initiation, treatment duration, and genomic testing).

### Methods and findings

The Carolina Breast Cancer Study (CBCS) Phase 3 is a population-based, case-only cohort ($n = 2,998$, 50% black) of patients with invasive breast cancer diagnoses (2008 to 2013). We used latent class analysis (LCA) to group participants based on patterns of factors within 3 separate domains: socioeconomic status ("SES"), "care barriers," and "care use." These classes were evaluated in association with delayed diagnosis (approximated with stages III–IV at diagnosis), delayed treatment initiation (more than 30 days between diagnosis and first treatment), prolonged treatment duration (time between first and last treatment–by treatment modality), and receipt of OncotypeDx genomic testing (evaluated among patients with early stage, ER+ (estrogen receptor-positive), HER2- (human epidermal growth factor receptor 2-negative) disease). Associations were evaluated using adjusted linear-risk regression to estimate relative frequency differences (RFDs) with 95% confidence intervals (CIs). Delayed diagnosis models were adjusted for age; delayed and prolonged treatment models were adjusted for age and tumor size, stage, and grade at diagnosis; and OncotypeDx models were adjusted for age and tumor size and grade.

data that support the findings of this study are available upon submission of a letter of intent and approval from the Carolina Breast Study Steering Committee (https://ciphr.unc.edu/cbcs-loi-form.php) and IRB approval. The data are not publicly available to protect privacy of study participants.

**Funding:** M.R.D. was supported by the National Cancer Institute's National Research Service Award sponsored by the Lineberger Comprehensive Cancer Center (https://unclineberger.org/outcomes/ccqtp/) at the University of North Carolina (T32 CA116339 to M.R.D.). This research was supported by a grant from UNC Lineberger Comprehensive Cancer Center, which is funded by the University Cancer Research Fund of North Carolina (https://unclineberger.org/ucrf/), the Susan G Komen Foundation (https://www.komen.org/) (OGUNC1202, OG22873776, SAC210102, TREND21686258; to M.A.T.), National Cancer Institute (https://www.cancer.gov/) (R01CA253450 to M.A.T.), the National Cancer Institute Specialized Program of Research Excellence (SPORE) in Breast Cancer (NIH/NCI P50-CA058223 to M.A.T.), the Breast Cancer Research Foundation (HEI-23-003 to M.A.T.) (https://www.bcrf.org/), and the US Department of Defense (https://www.defense.gov/) (HT94252310235 to M.A.T.). This research recruited participants and/or obtained data with the assistance of Rapid Case Ascertainment, a collaboration between the North Carolina Central Cancer Registry and UNC Lineberger Comprehensive Cancer Center. Rapid Case Ascertainment is supported by a grant from the NCI of the NIH (grant no. P30CA016086). The Pathology Services Core is supported in part by NCI of the NIH Center Core Support Grant (P30CA016080) and the UNC-CH University Cancer Research Fund. The funders had no role in study design, data collection and analysis, decision to publish, or preparation of the manuscript.

**Competing interests:** M.L.R. received consulting fees from Concert Genetics and National Committee for Quality Assurance, unrelated to the submitted work.

**Abbreviations:** AIC, Akaike information criterion; AJCC, American Joint Committee on Cancer; BIC, Bayesian information criterion; CBCS, Carolina Breast Cancer Study; CI, confidence interval; EM, expectation-maximization; ER+, estrogen receptor positive; HER2, human epidermal growth factor receptor 2-negative; HR, hazard ratio; LCA, latent class analysis; RFD, relative frequency difference; RUCA, rural-urban commuting area; SDM, shared decision-making; SES, socioeconomic status; USPSTF, United States Preventive Service Task Force.

Overall, 18% of CBCS participants had late stage/delayed diagnosis, 35% had delayed treatment initiation, 48% had prolonged treatment duration, and 62% were not OncotypeDx tested. Black women had higher prevalence for each outcome. We identified 3 latent classes for SES ("high SES," "moderate SES," and "low SES"), 2 classes for care barriers ("few barriers," "more barriers"), and 5 classes for care use ("short travel/high preventive care," "short travel/low preventive care," "medium travel," "variable travel," and "long travel") in which travel is defined by estimated road driving time. Low SES and more barriers to care were associated with greater frequency of delayed diagnosis ($RFD_{adj}$ = 5.5%, 95% CI [2.4, 8.5]; $RFD_{adj}$ = 6.7%, 95% CI [2.8,10.7], respectively) and prolonged treatment ($RFD_{adj}$ = 9.7%, 95% CI [4.8 to 14.6]; $RFD_{adj}$ = 7.3%, 95% CI [2.4 to 12.2], respectively). Variable travel (short travel to diagnosis but long travel to surgery) was associated with delayed treatment in the entire study population ($RFD_{adj}$ = 10.7%, 95% CI [2.7 to 18.8]) compared to the short travel, high use referent group. Long travel to both diagnosis and surgery was associated with delayed treatment only among black women. The main limitations of this work were inability to make inferences about causal effects of individual variables that formed the latent classes, reliance on self-reported socioeconomic and healthcare history information, and generalizability outside of North Carolina, United States of America.

## Conclusions

Black patients face more frequent delays throughout the care continuum, likely stemming from different types of access barriers at key junctures. Improving breast cancer care access will require intervention on multiple aspects of SES and healthcare access.

## Author summary

### Why was this study done?

- Delays in breast cancer care can lead to worse outcomes and there are documented racial disparities in who receives timely care.

- Few prior studies of breast cancer treatment timeliness have attempted to integrate the complex set of factors that shape patients' access to care.

### What did the researchers do and find?

- We used latent class methods to generate composite measures of healthcare access that include socioeconomic status (SES), care barriers, and care use domains.

- These composite variables were evaluated in association with multiple markers of breast cancer care delays (diagnosis, treatment initiation, treatment duration).

- Patients with low SES and more barriers to care had greater frequency of delayed diagnosis and prolonged treatment duration.

- Longer driving time to care was associated with greater frequency of delayed treatment, particularly among black patients.

**What do these findings mean?**

- Different sets of healthcare access factors were associated with different markers of delay, highlighting how different barriers may be more salient at different stages of cancer care.

- Black women had greater frequency of each type of care delay, which suggest that eliminating racial disparities in breast cancer care will require intervening on multiple dimensions of access.

- Health system efforts to assess patient risk of delayed care (and refer high-risk patients to supportive services such as patient navigation) can be strengthened by consideration of multiple access factors rather than any one factor alone.

- A limitation of composite measures of healthcare access (e.g., SES) is that they do not assess the effect of specific factors (e.g., income) and further research is needed to understand which factors should be prioritized for intervention planning.

## Introduction

Care delays occur at multiple points in the breast cancer care continuum, beginning with diagnosis. In the United States of America (USA), about one-third of breast cancer cases are regional or distant (spread beyond the breast) at diagnosis, for which survival is lower relative to localized [1–3]. After diagnosis, at least 30% of patients wait 1 month or more to begin treatment, and median wait times to surgery have increased over time, with one study estimating a shift from 18 to 28 days between 2004 and 2013 [4–7]. These are meaningful differences given that surgical delays (as first-course treatment or following neo-adjuvant chemotherapy) as short as 30 days are associated with lower survival and higher probability of recurrence [8–11]. Prolonged duration of adjuvant chemotherapy and other postoperative treatment regimens, while less studied, can also reflect care delays [12,13]. For example, in prior work from the Carolina Breast Cancer Study (CBCS), we found that participants with lower socioeconomic status (SES) and more barriers to care had greater frequency of prolonged treatment [14]. Delays at each stage of care are associated with race, with substantial evidence demonstrating that black women and women with low SES wait longer for surgery, chemotherapy, radiotherapy, and hormone therapy [6,7,13,15,16]. Removing the barriers to timely healthcare is necessary to improve outcomes and equity for people with breast cancer.

While many studies have examined potential causes of these separate timeliness measures, few studies have evaluated barriers across the continuum collectively. Prior studies have also been limited in their focus on particular barriers. Financial burden and insurance status have been well studied as root causes of care delays, but the magnitude of the burden associated with each may differ throughout the care process [17–22]. Other barriers to care are less studied, and additional research is needed to understand their impact on care delays and how that impact may vary across the care continuum.

The present study expands on previous work by assessing additional points along the cancer care continuum (diagnosis and genomic testing) as well as considering additional access factors (such as travel burden and history of preventive care) that predict delays. Overall, the objectives of this study are (1) to estimate associations between latent measures of healthcare

access with timeliness of diagnosis, treatment initiation, and treatment completion; (2) to esti-mate associations between healthcare access with receipt of the OncotypeDx genomic test; and (3) to describe differences in care timeliness between black and non-black patients.

## Materials and methods

### Study population

The CBCS Phase 3 is a population-based, case-only cohort of 2,998 women diagnosed with breast cancer in central and eastern North Carolina, USA. Patients with a first primary breast cancer diagnosis between 2008 and 2013 were eligible for inclusion. The CBCS used a random-ized sampling strategy with oversampling for black and younger women, such that half of the study population is black and half is aged 50 years or younger. Detailed methodology for the CBCS has been described previously [23,24]. The present study integrates data from partici-pant surveys, medical records, and pathology reports. Study nurses administered in-home sur-veys to participants approximately 5 months after their breast cancer diagnosis (2008 to 2013). The present analysis is reported as per the Strengthening the Reporting of Observational Stud-ies in Epidemiology (STROBE) guideline (S1 Checklist). All study protocols were approved by the Institutional Review Board of the University of North Carolina at Chapel Hill (Reference ID 413140).

### Exposure assessment

For this study, we used latent class analysis (LCA) to identify groupings of participants based on shared characteristics across three theoretical access domains: "socioeconomic status (SES)," "care barriers," and "care use." We constructed separate latent class models for each of these thematic domains. LCA methods are further described in the statistical analysis subsec-tion. Measurement of the specific factors that compose each domain are described as follows:

**SES factors.** SES variables were defined as a binary or multinomial categorical variable: household income (<$15,000, $15,000 to 50,000, or $50,000+), education (<high school, high school or equivalent, or at least some college), nativity (US or foreign born), job type (farmer/ service worker/other laborer, craftworker/factory worker/mechanic, clerical worker/sales/ technician, or professional/administrative/executive), and marital status (married or not married).

**Care barrier factors.** Participants reported whether they had experienced financial barri-ers to seeing a doctor, transportation barriers to seeing a doctor, or job loss after their breast cancer diagnosis (as loss of insurance or income would be a barrier to care). Participants also reported their insurance type, which we classified as insured (private, Medicare, or Medicaid) or not insured. The final variable was urban/rural status, determined based on participant home addresses. We used data from the National Cancer Institute's census tract-level SES and Rurality Database, which defines urban census tracts as those with rural-urban commuting area (RUCA) codes of 1.0, 1.1, 2.0, 2.1, 3.0, 4.1, 5.1, 7.1, 8.1, and 10.1, and defines all other codes as rural [25,26].

**Care use factors.** The third domain included factors related to observed use of healthcare. We included 3 indicators of pre-diagnostic preventive healthcare history. One variable was a self-reported measure of screening mammography history (defined from after age 40 years until 2 years prior to diagnosis), categorized as screening-adherent (at least 1 mammogram every 2 years) or under-screened (<0.5 per year), based on updated United States Preventive Service Task Force (USPSTF) guidelines which recommend biennial breast screening for most women aged 40 to 74 years [27]. We also defined self-reported "regular care," which reflects whether CBCS participants self-reported having source of regular healthcare in the 10 years

preceding their diagnosis. Those who relied on primary or specialty care were classified as having regular care, while those who indicated emergency or no source of care were classified as not having regular care. Construction and validation of these 2 variables are described in more detail in a previous study [28]. The third indicator was mode of detection, where participants reported if initial detection of their cancer was a routine screening test or a lump noticed by self/partner/doctor. We also included 2 measures of estimated road driving times between participant home and chosen healthcare facility. We estimated travel for confirmed diagnosis and surgery (but not other treatment endpoints) because almost all participants had records of visits for these forms of care, whereas not everyone received other forms of treatment (e.g., chemotherapy and radiotherapy). We note that measures of travel time for diagnosis and surgery are captured at different time points, but here we include them as measures of a latent characteristic indicative of distance-related patterns of travel to cancer care experienced by patients in this study.

## Demographic and clinical covariates

Race was measured by participant self-classification, consistent with our interpretation of race as a social construct rather than a measure of genetic ancestry [29]. For this analysis, participants were grouped as black ($n = 1,495$) or non-black ($n = 1,503$); 95% of the latter group are non-Hispanic white women. Tumor characteristics at diagnosis, including American Joint Committee on Cancer (AJCC) stage, tumor size, and grade were obtained from participant medical records. These tumor factors were binarized for our analyses: stage was categorized as I or II (early) versus III or IV (late), grade was 1 or 2 (low) versus 3 (high), and tumor size was <5 versus 5+ cm. Estrogen receptor (ER) status and human epidermal growth factor receptor 2 (HER2) status were obtained from pathology reports.

## Outcome assessment

We evaluated 4 healthcare timeliness and access outcomes based on treatment information from abstracted medical records and pathology reports. The first was diagnostic timeliness, which has been shown to correlate with earlier tumor stage at diagnosis [30,31]. We defined a late diagnosis as a stage III or IV tumor—characterized by larger and more advanced disease than if the cancer was diagnosed earlier.

The second outcome was treatment initiation timeliness, which was defined as greater than 30 days between confirmed diagnosis and initiation of first treatment (surgery, chemotherapy, or radiation). Although some studies have concluded there is a "safe window" of up to 45 to 60 days of delay [32–34], other studies have found decrements in survival beyond a 30-day wait to treatment, including a 2020 meta-analysis which found greater breast cancer mortality for delayed surgery (hazard ratio (HR) = 1.08, 95% CI [1.03, 1.13]) and chemotherapy (HR = 1.09, 95% CI [1.07, 1.11]) [8–11]. Therefore, in consideration of potential decrements in survival, as well as the burden of anxiety experienced by many patients during their wait to treatment, we elected to use a 30-day cutoff for delayed treatment [22,35]. Given the relative underrepresentation of stage IV patients in breast cancer health services research, we elected to include patients of all stages in this analysis. Sensitivity analyses showed that results were consistent when restricting to stages I–III patients (S3 Table).

The third outcome was prolonged treatment, defined by the amount of time between treatment initiation and completion. While less studied than treatment initiation, one study found that time greater than 38 weeks from diagnosis to treatment completion was associated with decrements in survival [36]. We used American Cancer Society treatment length recommendations to define "prolonged treatment" for each treatment modality [37,38]. Consistent with

prior CBCS papers which assessed treatment duration, treatment modalities were based on receipt of surgery, radiation, and/or chemotherapy [13,39]. Other treatments that may require multiyear administration (e.g., endocrine therapy) were not considered. For those who received only surgery and radiation, prolonged treatment was defined as >2 months between first and last treatment; for surgery and chemotherapy, prolonged treatment was >6 months between first and last treatment; and for surgery, radiation, and chemotherapy, prolonged treatment was >8 months between first and last treatment [37–39]. Participants who had surgery only ($N = 376$) were excluded from this analysis because their treatment duration was only 1 day. We also excluded stage IV cancers from the regression models ($N = 112$) because these patients often receive systemic therapy (e.g., trastuzumab) lasting beyond 1 year [40]. Sensitivity analyses were performed with stratification by ER and HER2 status since treatment pathways (and thus treatment duration) can also differ by breast cancer subtype.

Composite outcomes for delayed diagnosis, delayed treatment, and prolonged treatment were also generated among patients who were eligible for all 3 outcomes. Patients were classified as having 0, 1, 2, or all 3 of these outcomes; the relative frequency of each group was presented in Sankey flow diagrams.

The fourth outcome was receipt (yes/no) of the OncotypeDx genomic test among participants with Stage I/II, ER+, HER2- disease ($N = 1,615$), ascertained from pathology reports. OncotypeDx is a 21-gene assay of breast cancer tissue and is recommended for patients with estrogen receptor positive (ER+), human epidermal growth factor receptor negative (HER2-) breast cancers [41,42]. This test estimates risk of recurrence and is primarily used to help inform chemotherapy decisions; however, although usually covered by insurance, less than half of eligible women receive OncotypeDx [43,44].

### Statistical analysis

Constructs such as healthcare access are multifaceted and are shaped by the simultaneous experience of multiple characteristics (e.g., income, education, insurance status) [45]. We assessed patterns of these characteristics using LCA, an iterative method for identifying unmeasured (or latent) group membership in populations. In LCA, multiple indicator variables are assessed together to identify latent groupings (classes) based on shared characteristics of the observed indicator variables [46]. This method uses an expectation-maximization (EM) algorithm to determine class membership probabilities by maximizing the log-likelihood function [47]. We determined a priori which indicator variables were included into the latent class model, and we assessed a series of models with different numbers of classes (range of 2 to 10). Following best practice for LCA, we chose the optimal class number based on both interpretability and model fit [46]. We evaluated model fit by comparing Akaike information criterion (AIC) and Bayesian information criterion (BIC), model entropy, and the average probabilities of class membership.

We described the distribution of each latent class exposure and healthcare outcome. Exposure-outcome relationships were assessed using generalized linear models and estimation of relative frequency differences (RFDs) with 95% confidence intervals (CIs). Adjusted RFDs are indicated by subscript ($RFD_{adj}$). Covariate adjustment sets were determined a priori for each outcome based on clinical factors which may correlate with latent class membership and treatment timeliness. Late diagnosis models were adjusted for age; delayed and prolonged treatment models were adjusted for age and tumor size, stage, and grade at diagnosis; and OncotypeDx models were adjusted for age, and tumor size and grade (only evaluated among early-stage patients). All regression models are presented overall and race-stratified (black versus non-black). Complete case analysis was used given there was no missingness for exposures

and <2% missingness for outcomes ($N = 3$ for delayed diagnosis, $N = 3$ for delayed treatment, $N = 54$ for prolonged treatment, and $N = 20$ for OncotypeDx). We also report prevalence estimates of care delays for each potential combination of latent class membership across all 3 domains. Finally, although we used latent class modeling to capture multidimensional characteristics of healthcare access, we also wanted to compare these findings to single-variable approaches (i.e., only income or only variable travel). Therefore, we evaluated each care outcome in relation to household income (<15 k, 15–50 k, 50 k+) and variable travel (<30 min difference in travel time for diagnosis and surgery, 30 to 60 min difference, and 60+ min difference). Data analysis and visualization was performed SAS version 9.4 (SAS Institute; Cary, North Carolina, USA), R version 4.3.1 (R Foundation for Statistical Computing; Vienna, Austria), and MPlus version 8.0 (Los Angeles, California, USA).

## Results

### Outcome distributions

Clinical and demographic characteristics, including details on SES, care barriers, and care use of the study population are reported in S1 and S2 Tables. Care delays were more frequent at later stages of the cancer care continuum. Among the total 2,998 CBCS participants, 18% had a delayed diagnosis, 35% had delayed treatment initiation, and 48% had prolonged treatment duration (Fig 1). About three-quarters of participants had at least 1 delay, and about one-quarter had 2 or 3 delays. At each step, black women were more likely to experience delays than

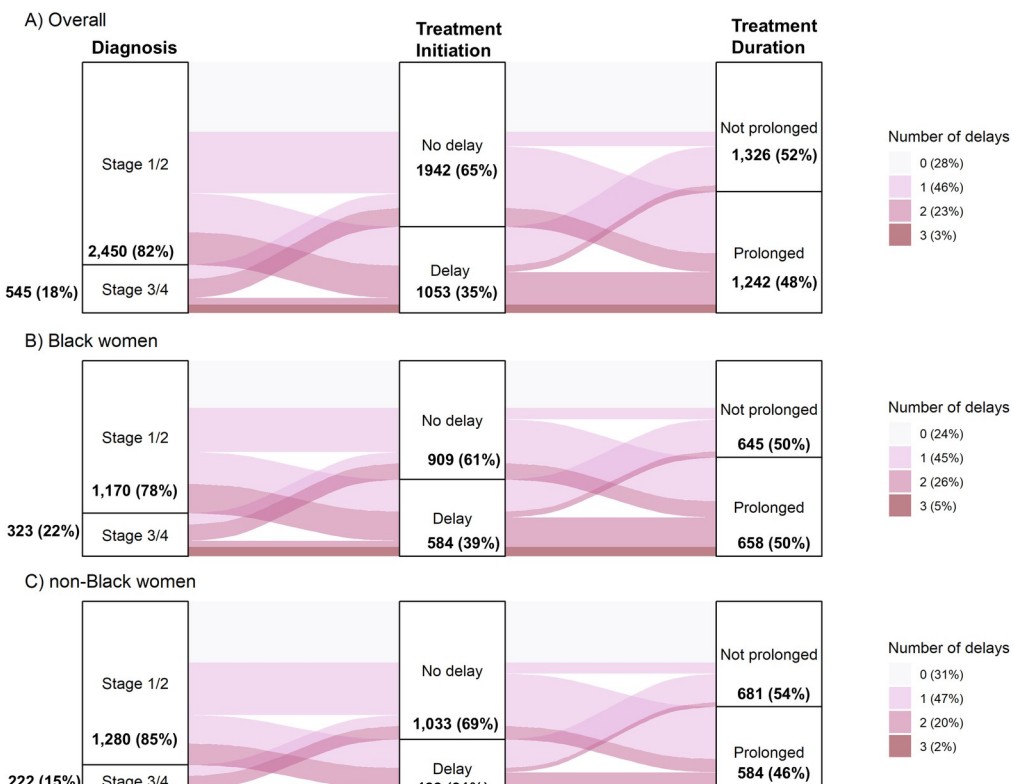

**Fig 1. Sankey flow diagram of potential delay points among patients receiving chemotherapy and/or radiation.** Care timeliness was assessed for breast cancer diagnosis (stage I/II vs. III/IV), treatment initiation (no delay/delay), and treatment duration (not prolonged/prolonged). The vertical height of each pathway indicates the relative frequency of each pathway. Color shading indicates the total number delays (minimum = 0, maximum = 3). Results are presented (A) for the overall study population, (B) for black women, and (C) for non-black women.

**Table 1. Distribution of latent class membership and care outcomes by race (*N* = 2,998).**

| | Overall | non-black | Black | Contrast |
|---|---|---|---|---|
| | *N* = 2,998 | *N* = 1,503 | *N* = 1,495 | |
| | *N* (%) | *N* (%) | *N* (%) | RFD (95% CI)* |
| Latent class domains | | | | |
| **Socioeconomic status (SES)** | | | | |
| High SES | 1,100 (37%) | 757 (50%) | 343 (23%) | ref |
| Moderate SES | 726 (24%) | 257 (17%) | 469 (31%) | 14.3 (11.2, 17.3) |
| Low SES | 1,172 (39%) | 489 (33%) | 683 (46%) | 13.2 (9.7, 16.6) |
| **Care barriers** | | | | |
| Fewer barriers | 2,478 (83%) | 1,346 (90%) | 1,132 (76%) | ref |
| More barriers | 520 (17%) | 157 (10%) | 363 (24%) | 13.8 (11.2, 16.5) |
| **Care use** | | | | |
| Short travel, high use | 1,750 (58%) | 868 (58%) | 882 (59%) | ref |
| Short travel, low use | 415 (14%) | 136 (9.0%) | 279 (19%) | 9.6 (7.2, 12.1) |
| Medium travel | 551 (18%) | 336 (22%) | 215 (14%) | −8.0 (−10.7, −5.2) |
| Variable travel | 164 (5.5%) | 100 (6.7%) | 64 (4.3%) | −2.4 (−4.0, −0.7) |
| Long travel | 118 (3.9%) | 63 (4.2%) | 55 (3.7%) | −0.5 (−1.9, 0.9) |
| Care outcomes | | | | |
| **Diagnosis** | | | | |
| Not delayed | 2,450 (82%) | 1,280 (85%) | 1,170 (78%) | ref |
| Delayed | 545 (18%) | 222 (15%) | 323 (22%) | 6.9 (4.1, 9.6) |
| **Treatment initiation** | | | | |
| Not delayed | 1,942 (65%) | 1,033 (69%) | 909 (61%) | ref |
| Delayed | 1,053 (35%) | 469 (31%) | 584 (39%) | 7.9 (4.5, 11.3) |
| **Treatment duration** | | | | |
| Not prolonged | 1,326 (52%) | 681 (54%) | 645 (50%) | ref |
| Prolonged | 1,242 (48%) | 584 (46%) | 658 (50%) | 4.3 (0.5, 8.2) |
| **Receipt of OncotypeDx** | | | | |
| Tested | 609 (38%) | 401 (43%) | 208 (31%) | ref |
| Not tested | 1,006 (62%) | 537 (57%) | 469 (69%) | 12.2 (7.3–16.7) |

* Relative frequency difference (RFD, expressed as a percentage difference on the absolute scale) and 95% confidence interval, comparing frequencies of latent class domains between black women and non-black women (referent group).

Latent variables were defined for SES (income, education, country of birth, job type, and marital status), care barriers (insurance, urban/rural status, job loss, self-reported financial barriers to care, self-reported transportation barriers to care), and care use (pre-diagnostic regular care, breast cancer screening, mode of initial cancer detection (mammogram vs. noticed lump), and travel (based on estimated driving time) to diagnosis and surgery).

OncotypeDx is a genomic test of breast cancer tissue used to assess risk of recurrence and inform chemotherapy treatment decisions.

non-black women (Table 1), including diagnostic delay (22% versus 15%; RFD = 6.9%, 95% CI [4.1, 9.6]), treatment initiation delay (39% versus 31%; RFD = 7.9%, 95% CI [4.5, 11.3]), and prolonged treatment (50% versus 46%; RFD = 4.3%, 95% CI [0.5, 8.2]). We observed a similar pattern for receipt of OncotypeDx: 62% of the cohort had no evidence of testing, and black women were more likely to be untested compared to non-black women (69% versus 57%; RFD = 12.2%, 95% CI [7.3, 16.7]).

## Latent class membership

S1 Fig shows the posterior probabilities for input characteristics given their latent class membership for each of the access domains. For SES, we selected a 3-class solution (which had

slightly better model fit compared to a 2-class solution, and much better model fit compared to a 4-class solution). In this model, $N = 1,100$ out of 2,998 participants (37%) were "high SES" (greater probabilities of high income, education, and other advantageous SES factors), $N = 726$ (39%) were "low SES," and $N = 1,172$ (24%) were "moderate SES" (high probability of college education but low probabilities of other advantageous characteristics) (Table 1). Relative to the high SES group, black women were more frequently classified as low SES (31% versus 23%; RFD = 14.3%, 95% CI [11.2, 17.3]) or moderate SES group (46% versus 23%; RFD = 13.2%, 95% CI [9.7, 16.6]). For care barriers, a 2-class solution had substantially better model fit compared to all other solutions (Table 1). There were $N = 520$ (17%) participants classified as having "more barriers" (greater probabilities of barriers such as lacking insurance), and the remaining $N = 2,450$ (82%) were classified as having "few barriers." Black women more frequently were classified as having more barriers compared to non-black women (24% versus 10%; RFD = 13.8%, 95% CI [11.2, 16.5]).

For care use, a 5-class solution had better fit than 2-, 3-, or 4-class models (S1 Fig). Beyond 5-classes, there were unstable groups with fewer than 50 participants. In the selected model, participants were differentiated by their travel to care and self-reported preventive healthcare utilization. Most participants had short travel to care (<30 min driving time by road) and were further subdivided by their preventive healthcare use. Thus, two of the classes were "short travel, high use" ($N = 1,750$ out of 2,998 participants; 58%) and "short travel, low use" ($N = 415$; 14%) (Table 1). The remaining participants were classified as "medium travel" ($N = 551$; 18%, characterized by higher probabilities of 30 to 60 min of travel), "long travel" ($N = 118$; 4%, characterized higher probabilities of 60+ min of travel), and "variable travel" ($N = 164$; 6%, characterized by higher probabilities of shorter travel to diagnosis and longer travel to surgery). Black women were more frequently classified as short travel, low use (RFD = 9.6%, 95% CI [7.2, 12.1]) compared to non-black women (Table 1). Overall, there were 30 possible combinations of latent class membership across the domains of SES, care barriers, and care use (S4 Table). Each potential combination has at least 3 patients, and 19 out of 30 combinations had at least 30 patients, but there were also correlations between domains. For example, low SES patients were more likely to have more care barriers.

## Associations between latent classes and outcomes

Low SES participants had more delayed diagnoses (19% versus 15%; $RFD_{adj}$ = 5.5%, 95% CI [2.4, 8.5]), as did moderate SES participants (21% versus 15%; $RFD_{adj}$ = 6.0%, 95% CI [2.4, 9.5]) compared to the high SES group (Fig 2). The magnitude of these differences was larger for black women, especially when contrasting late-stage diagnosis among moderate SES participants versus high SES participants, where the $RFD_{adj}$ was 9.3% (95% CI [4.0, 14.7]) for black women and null (−0.8%, 95% CI [−5.7, 4.0]) for non-black women. Among the 2 barriers classes, participants with more healthcare barriers had more frequent diagnosis delays (24% versus 17%; $RFD_{adj}$ = 6.7%, 95% CI [2.8, 10.7]), with similar patterns for black and non-black women (Fig 2). Among the 5 care use classes, participants in the low travel, low use group had substantially more diagnosis delays (33% versus 15%; $RFD_{adj}$ = 17.4%, 95% CI [12.6, 22.2]) compared to the short travel, high use referent group.

The second outcome of interest was treatment initiation delay. None of the SES or care barriers latent class models were associated with treatment delay (Fig 3). For care use classes, participants with variable travel had greater frequency of treatment delay (46% versus 35%; $RFD_{adj}$ = 10.7%, 95% CI [2.7, 18.8]) compared to the short travel, high use group (Fig 3). This association was also observed in the race-stratified models; both black and non-black women with variable travel had more treatment delays. Long travel participants showed no association

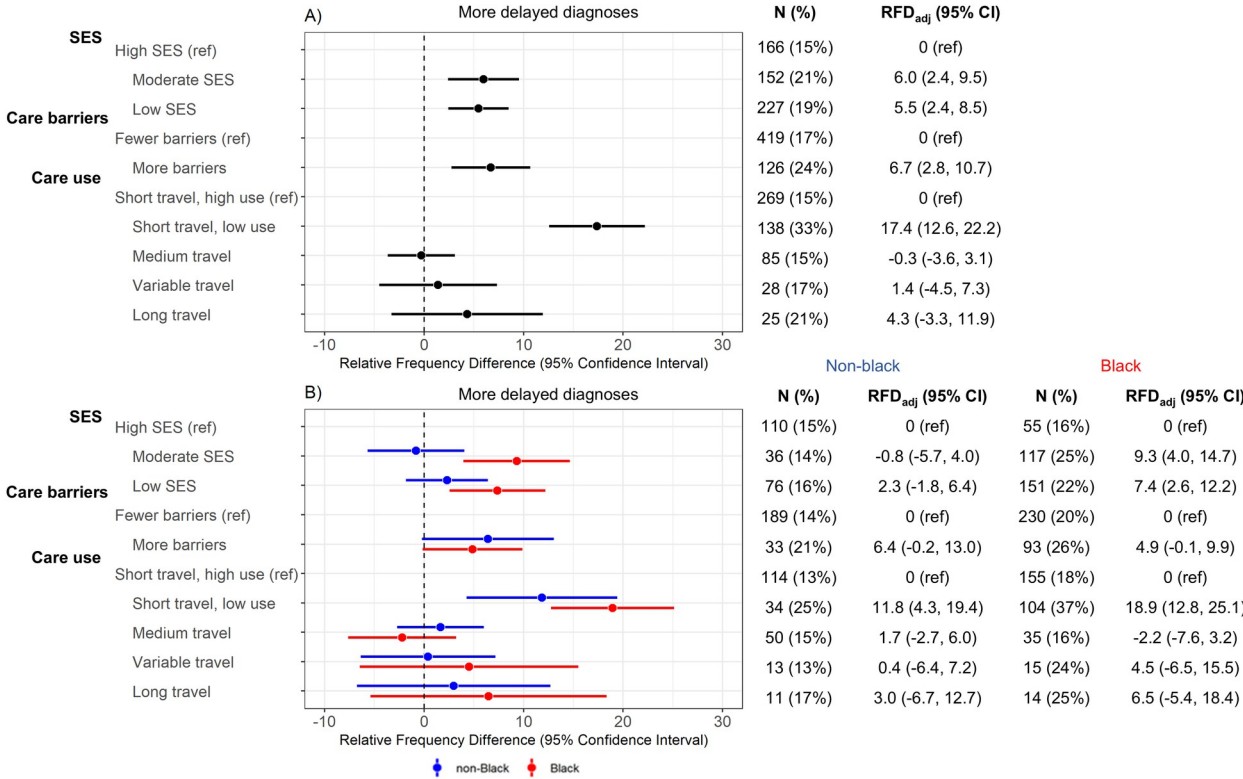

**Fig 2. Linear-risk regression models for delayed diagnosis (N = 2,995).** Delayed diagnosis was assessed in relation to latent variables defined for SES (income, education, country of birth, job type, and marital status), care barriers (insurance, urban/rural status, job loss, self-reported financial barriers to care, self-reported transportation barriers to care), and care use (pre-diagnostic regular care, breast cancer screening, mode of initial cancer detection (mammogram vs. noticed lump), and travel (based on estimated driving time) to diagnosis and surgery). Frequency and percentage of delayed diagnosis is reported in "n(%)" columns. Contrast estimates are RFDs and 95% CIs, which compare frequency of delayed diagnosis in a given latent class with the indicated reference group. Results are presented (A) overall and (B) stratified by race; all models are age-adjusted. CI, confidence interval; RFD, relative frequency difference; SES, socioeconomic status.

with treatment delay in the overall population; however, we observed divergent results in race-stratified models. Black women with long travel had more treatment delays (RFD$_{adj}$ = 10.0%, 95% CI [−3.3, 23.2]) while non-black women with long travel had fewer treatment delays (RFD$_{adj}$ = −6.5%, 95% CI [−17.2, 4.2]). Patterns were similar in sensitivity analyses using a 45-day (rather than 30-day) cutoff for delayed treatment (S2 Fig).

The third outcome of interest was prolonged treatment duration, which showed similar patterns to delayed diagnosis models. Compared to high SES, there were greater frequencies of prolonged treatment among the low SES (51% versus 42%; RFD$_{adj}$ = 9.7%, 95% CI [4.8, 14.6]) and moderate SES (53% versus 42%; RFD$_{adj}$ = 10.1%, 95% CI [5.7, 14.5]) groups (Fig 4). There was also greater frequency of prolonged treatment among participants with more care barriers compared to few barriers (54% versus 47%; RFD$_{adj}$ = 7.3%, 95% CI [2.4, 12.2]). These associations were similar for both black and non-black women (Fig 4). None of the utilization latent classes were associated with prolonged treatment. In sensitivity analyses stratified by ER+/ER- and HER2+/HER2- disease, the variable travel group was associated with more frequent prolonged treatment (relative to the short travel, high use referent group) only among participants with ER⁻ (RFD$_{adj}$ = 22.4%, 95% CI [4.5, 40.3]) and HER2⁺ disease (RFD$_{adj}$ = 22.8%, 95% CI [−0.5, −46.2]); these associations were null for ER⁺ and HER2⁻ (S3 Fig). For the remaining comparisons, RFDs were similar by ER and HER2 status.

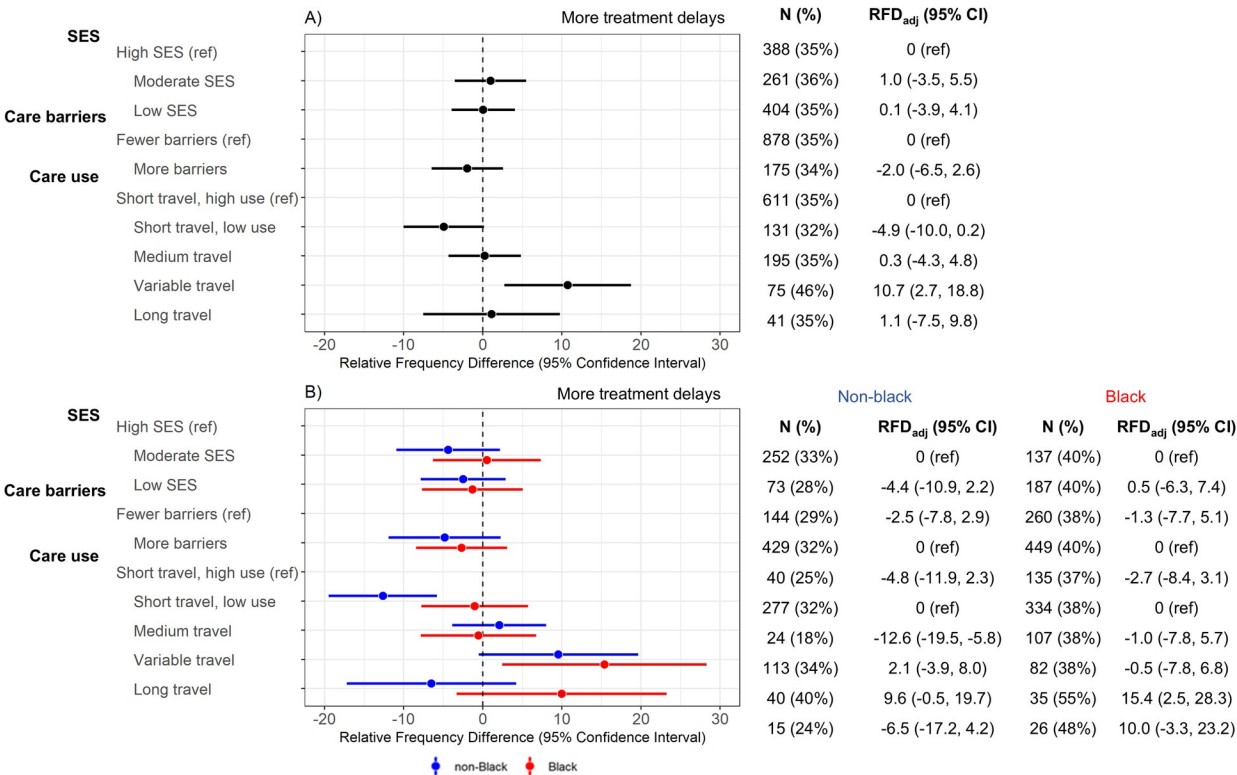

**Fig 3. Linear-risk regression models for delayed treatment (*N* = 2,947).** Delayed treatment was assessed in relation to latent class variables were defined for SES (income, education, country of birth, job type, and marital status), care barriers (insurance, urban/rural status, job loss, self-reported financial barriers to care, self-reported transportation barriers to care), and care use (pre-diagnostic regular care, breast cancer screening, mode of initial cancer detection (mammogram vs. noticed lump), and travel (based on estimated driving time) to diagnosis and surgery). Frequency and percentage of delayed diagnosis is reported in "n(%)" columns. Contrast estimates are RFDs and 95% CIs, which compare frequency of delayed treatment for a given latent class with the indicated reference group. Results are presented (A) overall and (B) race-stratified; models are adjusted for age, stage, size, and grade at diagnosis. CI, confidence interval; RFD, relative frequency difference; SES, socioeconomic status.

We then evaluated the outcomes of delayed diagnosis, treatment delay, and prolonged treatment in relation to patterns of latent class membership for all 3 domains (Fig 5). The prevalence of delayed diagnosis was highest in low travel, low use classes, particularly for patients who also had more care barriers: there were 18 out of 39 patients (46%) with delayed diagnosis among moderate SES, more barriers, and low travel, low use patients; and 44 out of 118 (37%) among low SES, more barriers, and low travel, low use patients. Delayed treatment was most prevalent among patients with variable or long travel, and had low or moderate SES. Prolonged treatment was associated with lower SES and more barriers groups; the highest prevalence (8 out of 12, 67%) was observed for the moderate SES, few barriers, and long travel group.

The fourth outcome, receipt of the OncotypeDx test among eligible participants (stage I–II, ER+, HER2- disease), was most strongly associated with SES. Non-receipt of OncotypeDx was more common among participants with low SES (66% versus 58%; $RFD_{adj}$ = 7.1%, 95% CI [1.5, 12.6]) and moderate SES (64% versus 58%; $RFD_{adj}$ = 5.1%, 95% CI [−1.1, 11.3]) compared to high SES (S4 Fig). These associations were attenuated somewhat when stratified by race. The barriers and care use latent classes were not associated with OncotypeDx.

Sensitivity analyses were performed to assess the potential impact of additional exclusion criteria. Effect estimates were similar in sensitivity analyses with black and white patients only

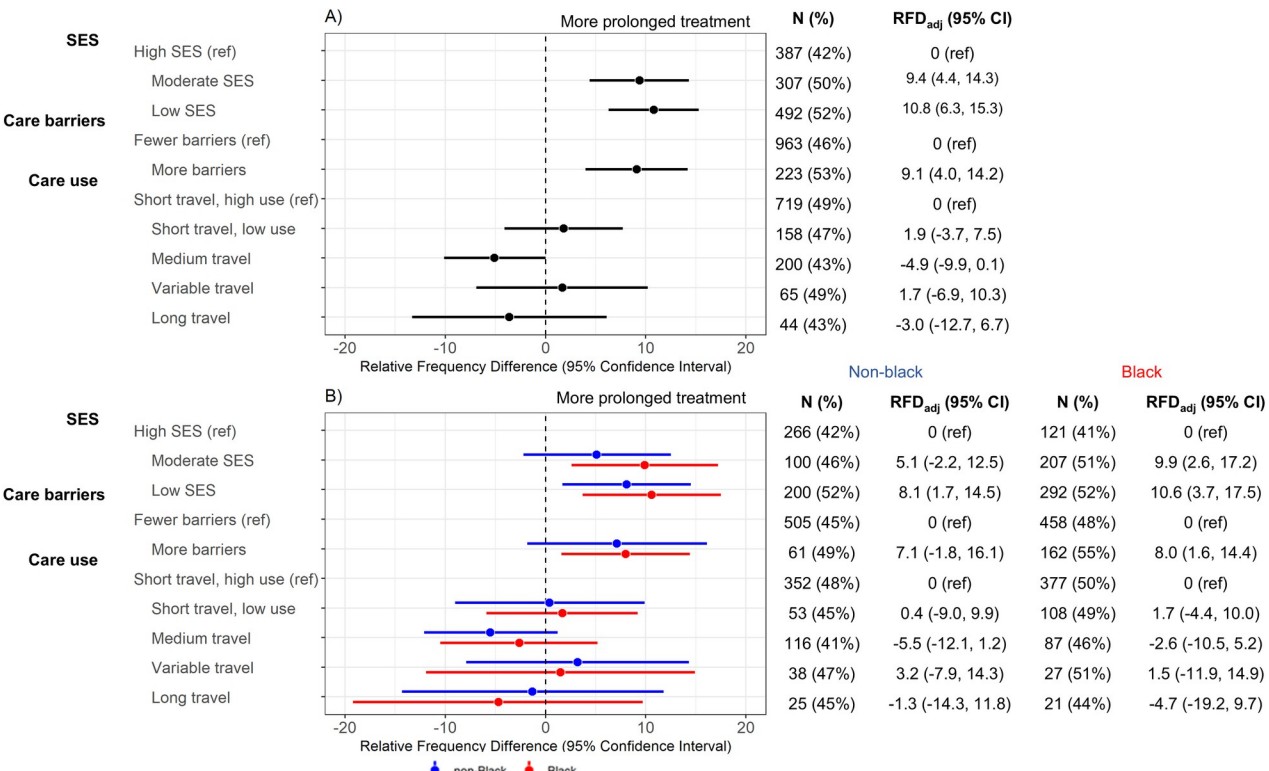

**Fig 4. Linear-risk regression models for prolonged treatment duration with surgery-only patients excluded (*N* = 2,469).** Prolonged treatment was assessed in relation to latent variables Latent variables were defined for SES (income, education, country of birth, job type, and marital status), care barriers (insurance, urban/rural status, job loss, self-reported financial barriers to care, self-reported transportation barriers to care), and care use (pre-diagnostic regular care, breast cancer screening, mode of initial cancer detection (mammogram vs. noticed lump), and travel (based on estimated driving time) to diagnosis and surgery). Frequency and percentage of delayed diagnosis is reported in "n(%)" columns. Contrast estimates are RFDs and 95% CIs, which compare frequency of prolonged treatment for a given latent class with the indicated reference group. Results are presented (A) overall and (B) race-stratified; models are adjusted for age, stage, size, and grade at diagnosis. CI, confidence interval; RFD, relative frequency difference; SES, socioeconomic status.

(*N* = 2,916) (S3 Table). Results for the delayed diagnosis and delayed treatment models were also consistent when restricting to stage I–III patients (*N* = 2,886) (S3 Table).

Table 2 shows adjusted linear-risk regression models for the 4 outcomes of interest, using 2 single variables (variable travel and income) as predictors rather than latent classes. More variable travel (greater difference in travel time to surgery versus diagnosis) was associated with treatment delay but no other outcome, consistent with our latent class results. In contrast, lower income was associated with all 4 outcomes (delayed diagnosis, delayed treatment, prolonged treatment, not OncotypeDx tested), while SES latent classes (which included income) were only associated with 2 of these outcomes: delayed diagnosis and prolonged treatment.

## Discussion

In this analysis of a racially diverse cohort of 2,998 women with breast cancer, we identified latent subgroupings of people according to SES, care barriers, and care use. Our objective was to estimate associations between latent class membership and 4 care outcomes: delayed diagnosis, treatment initiation delay, prolonged treatment duration, and not receiving the OncotypeDx test. Suboptimal outcomes were more frequent among black women at each step of the care continuum and different combinations of latent class domains associated with each

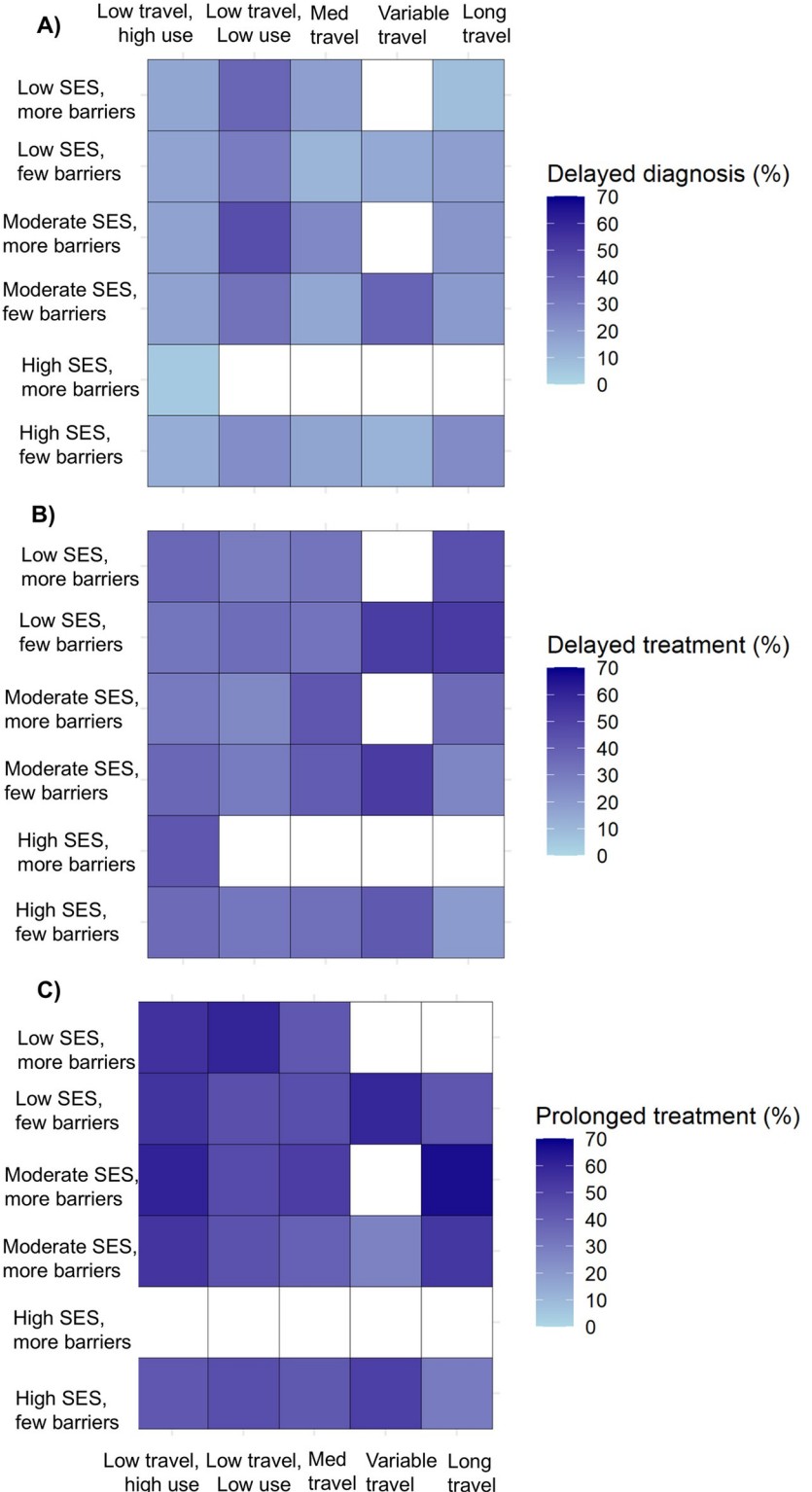

**Fig 5. Frequency of care delays according to latent class membership across all domains.** Latent variables were defined for SES (income, education, country of birth, job type, and marital status), care barriers (insurance, urban/rural status, job loss, self-reported financial barriers to care, self-reported transportation barriers to care), and care use (pre-diagnostic regular care, breast cancer screening, mode of initial cancer detection (mammogram vs. noticed lump), and travel (based on estimated driving time) to diagnosis and surgery). Frequency of care delays are shown according

to SES and care barriers (Y axis labels) and care use (X axis labels). Categories containing <10 patients were not assessed; these cells are shaded white. Results are presented for (A) delayed diagnosis, (B) delayed treatment, and (C) prolonged treatment. SES, socioeconomic status.

measure along the care continuum. Notably, latent classes of low SES, more barriers, and lower use of preventive care were associated with later diagnoses, and all but care use were associated with prolonged treatment duration. Participants with variable travel had greater frequency of delayed treatment initiation, while participants with consistently long travel had initiation delay only among black women.

Our findings highlight persistent racial disparities in breast cancer treatment timeliness regardless of the point along the continuum. Black women experience greater frequency of delayed and prolonged treatment, consistent with previous work from the CBCS [14]. Our study further highlighted disparities in OncotypeDx, where black women were again more likely to be underserved. Racial disparities in treatment timeliness are a persistent concern for cancer health equity, especially given that the adverse impact of treatment delay on survival may be largest among black women [6,48]. In addition, lower SES, more barriers, and low use of preventive care groups were each more prevalent among black participants. Thus, reducing disparities in treatment timeliness will likely demand policy solutions that address both healthcare specific factors (e.g., insurance and primary care availability) and the upstream economic and educational factors that create inequities in accessibility of healthcare.

We used a person-centered modeling approach to identify "latent" groupings of participants rather than causal modeling approach to estimate the effect of a single risk factor. An advantage of our approach was the ability to consider a conceptual array of factors that affect care timeliness, because individual factors may be misleading if they are inter-related with complex patterns. For example, we identified a latent class with short travel to care (which alone would indicate high access) yet also had low use of preventive care, indicating there were non-geographic distance barriers to care within this subgroup. Such methods may be useful for health systems to identify patients who are most vulnerable to care delays. Those patients could then be triaged for specific support services, particularly ones that organizations with constrained resources may not be able to provide to everyone. For example, targeted patient navigation services have been shown to reduce waiting times in breast and other cancers [49–

**Table 2. Isolated effect of 2 latent class inputs: variable travel and income.**

|  | Delayed diagnosis | Delayed treatment | Prolonged treatment | Not OncotypeDx tested |
|---|---|---|---|---|
|  | RFD (95% CI) | RFD (95% CI) | RFD (95% CI) | RFD (95% CI) |
| **Variable travel** |  |  |  |  |
| < = 30 min (ref) | 0 (ref) | 0 (ref) | 0 (ref) | 0 (ref) |
| 30–60 min | −0.8 (−6.7, 5.1) | 8.4 (0.6, 16.2) | 0.6 (−7.6, 8.9) | −5.7 (−16.1, 4.6) |
| 60+ min | 1.0 (−7.3, 9.3) | 11.4 (0.8, 21.9) | 3.1 (−7.7, 14.0) | −10.7 (−24.2, 2.8) |
| **Income** |  |  |  |  |
| $50,000+ (ref) | 0 (ref) | 0 (ref) | 0 (ref) | 0 (ref) |
| $15,000–$50,000 | 4.6 (1.6, 7.6) | −0.7 (−4.5, 3.1) | 7.4 (3.1, 11.8) | 5.1 (−0.4, 10.6) |
| <%15,000 | 10.2 (5.8, 14.6) | 6.3 (1.0, 11.5) | 14.9 (9.2, 20.6) | 16.2 (9.0, 23.5) |

Variable travel was defined based on the difference in road travel time between a patient's surgery and diagnosis healthcare visit. RFDs with 95% CIs were adjusted for age for delayed diagnosis, adjusted for age, stage, size, and grade at diagnosis for delayed treatment, prolonged treatment models were adjusted for age, stage, size, and grade, and OncotypeDx models were adjusted for age, tumor size, and grade at diagnosis.
CI, confidence interval; RFD, relative frequency difference.

51]. Expanding telehealth is another important step to alleviate barriers to timely care for vulnerable patients, although achieving wider implementation of telehealth in oncology will require improving provider capacity, reimbursement policies, and patient technology access [52,53]. One study in breast cancer found shared decision-making (SDM) was associated with fewer treatment interruptions for vulnerable patients, although overall there is fairly little study of SDM in relation to cancer treatment timeliness [54,55].

Another strength of our study was inclusion of estimated driving times between participant home and site of care. We identified multiple patterns of travel, including a "variable travel" group characterized by greater probability of shorter travel to diagnosis but longer travel to surgery, and a "long travel" group characterized by longer travel to both diagnosis and surgery. For both black and non-black women, the variable travel group was more likely to experience treatment initiation delay, potentially reflecting the vulnerability induced by a sudden transition from primary care in local facilities to specialized care in more distant facilities. This challenge is likely exacerbated in a predominantly rural state like North Carolina: one recent study found that geographic subregion in North Carolina was significantly associated with delays from diagnosis to first treatment [56]. To our knowledge, only 1 prior study assessed patient-level geo-coded data in relation to breast cancer treatment timeliness; a study from Vermont found that increased driving time was associated with longer wait times to chemotherapy [57]. Our study provides additional evidence that travel burden may affect treatment timeliness within racially diverse patient populations and more geographically dispersed states as well.

While the long travel group was associated with more treatment delay among black women, an inverse relationship was observed among non-black women. Unexpected associations of increased travel time with better quality of care (including more timely treatment) have been previously found for other cancers and are sometimes described as a "Travel Time Paradox" [58]. One hypothesis is that longer travel may reflect heterogenous ability to access healthcare: some people travel farther distances because they have the means to "shop" for the best providers, while others travel longer distances by necessity. Black women in the CBCS were more represented in disadvantaged latent classes, which may explain why they experience long travel as a burden, whereas non-black women did not. A similar conclusion was reached in a study of ovarian cancer, where the association of increased travel with non-receipt of guideline-recommended care was more pronounced among black women [59]. Alternatively, some studies have found that longer travel is associated with less timely care in urban areas, whereas in rural areas, longer travel correlated to more timely care [60,61].

A limitation of latent class analysis is that it precludes direct causal inferences about the effect of specific variables. Disaggregating latent classes into their component parts for analysis is still important because potential interventions will likely act on specific factors rather than entire classes. We also found some variables are highly influential when considered independently, suggesting that there may be some hierarchies of individual variables such that some factors are more influential and may even causally impact other factors within a given latent class. As an example, we showed that having a lower income was associated with all 3 care delay outcomes (diagnosis, initiation, duration) as well as not receiving the OncotypeDx test (also reported in a previous CBCS paper) [62]. However, the SES latent classes, which included income, were only associated with 2 outcomes: delayed diagnosis and prolonged treatment. This result may be explained by the greater complexity of barriers related to timeliness for multimodal treatment (e.g., surgery, chemotherapy, and radiation therapy), relative to a single test where one specific factor might drive access. Given that access consists of both entry and passage (which may reflect multiple points of re-entry) through the health system—each characterized by distinct barriers—it is unsurprising we found different sets of factors were associated with each care outcome [63,64]. Still, even when patterns of access factors are important

for a given outcome, it is difficult to intervene on a classification pattern unless the hierarchy of factors and interplay between them is better understood. To address these limitations, future studies may employ mixed-methods approaches to better elucidate the causal pathways and interactions between specific factors within latent classes. Additionally, advanced modeling techniques, such as structural equation modeling, could be utilized to untangle the complex relationships and hierarchies among influential variables. We did report frequency of care delays across combinations of latent class membership, but some combinations of latent classes had few patients which precluded inferences of causal interactions between the thematic domains.

There were additional limitations of our study. First, much of the information on SES, barriers, and care use characteristics were self-reported and may be defined by heterogenous categories. For example, screening mammography was self-reported and binarized as guideline adherent/non-adherent, but someone who was never screened for breast cancer may be more underserved than someone who is under-screened. Delay outcomes were also binarized for more interpretable effect estimates, and while avoiding any delay is preferable, there are substantial differences in implications of the length of such delays (e.g., 1 day versus 1 month). We were not able to account for certain case-mix factors such as pain or other symptoms that can affect timing of cancer treatment [65]. However, regression models controlled for other important clinical features at diagnosis, including tumor size, stage, and grade. Our latent class models also include a mix of factors that are time-varying. For example, the utilization model includes measures of preventive care utilization before diagnosis as well as travel time to surgery which is after diagnosis. Despite such limitations, we believe the value of such latent class constructs is to represent the totality of circumstances that determine accessibility of healthcare, rather than as evidence that latent classes are a cause of any outcome. In addition, at the time of this work, we were not able to consider community-level (e.g., poverty rate) or health system-level factors (e.g., hospital quality ratings). We hope to explore such factors in subsequent investigations. Finally, generalizability may be limited by patient geography (North Carolina, USA) and time of diagnosis (2008 to 2013), and experiences of delay may differ in countries with different health system organization (e.g., countries with universal public insurance).

In summary, there are multiple markers of timely access in the breast cancer continuum, each of which was associated with distinct sets of latent class indicators. Our results support a definition of healthcare access that integrates patient experience of multiple factors (e.g., income, insurance, travel burden) rather than any one factor alone, and health systems may consider these findings informative in designing algorithms for identifying and triaging patients for resource-limited patient navigation programs. Such interventions are particularly important given the complexity of the United States healthcare system and related challenges in access and affordability.

## Supporting information

**S1 Checklist. STROBE Checklist.**
(DOC)

**S1 Table. Clinical characteristics of the study population, overall and by race.** ER, estrogen receptor; HER2, human epidermal growth factor receptor 2.
(DOCX)

**S2 Table. Latent class inputs for SES, care barriers, and care use.** Counts and frequencies of variables used to form latent classes. Travel times were based on estimated driving distance

between patient residence and healthcare facilities. Breast cancer screening adherence was defined among patients age >45 based on receipt of at least 1 mammogram every 2 years.
(DOCX)

**S3 Table. Sensitivity analyses for restriction to black and white patients and restriction to stage I–III patients.** Cell values correspond to relative frequency differences (RFDs) and 95% confidence intervals. The main analysis column represents the CBCS population without exclusion criteria. The black and white patients only column ($N = 2,916$) represents a sensitivity analysis with Asian, American Indian, and other races excluded. The Stage I–III column ($N = 2,886$) represents a sensitivity analysis with stage IV patients excluded (not shown for prolonged treatment and OncotypeDx, as these analyses already excluded stage IV patients in the main analysis). RFDs for each outcome are compared between latent class categories, defined for SES (income, education, country of birth, job type, and marital status), care barriers (insurance, urban/rural status, job loss, self-reported financial barriers to care, self-reported transportation barriers to care), and care use (pre-diagnostic regular care, breast cancer screening, mode of initial cancer detection (mammogram vs. noticed lump), and travel (based on estimated driving time) to diagnosis and surgery).
(DOCX)

**S4 Table. Distribution of CBCS patients by cross classification of SES, care barriers, and care use.** Latent variables were defined for SES (income, education, country of birth, job type, and marital status), care barriers (insurance, urban/rural status, job loss, self-reported financial barriers to care, self-reported transportation barriers to care), and care use (pre-diagnostic regular care, breast cancer screening, mode of initial cancer detection (mammogram vs. noticed lump), and travel (based on estimated driving time) to diagnosis and surgery).
(DOCX)

**S1 Fig. Heatmap of latent class posterior probabilities.** Latent class distributions for domains of (A) SES (defined by income, education, country of birth, job type, and marital status), (B) care barriers (defined by insurance, urban/rural status, job loss, self-reported financial barriers to care, self-reported transportation barriers to care), and (C) care use (defined by (pre-diagnostic regular care, breast cancer screening, mode of initial cancer detection (mammogram vs. noticed lump), and travel (based on estimated driving time) to diagnosis and surgery). Cell values indicate posterior probabilities of individual items (left column) given latent class membership (top row).
(TIF)

**S2 Fig. Sensitivity analyses using a 45-day cut point for delayed treatment.** Delayed treatment was assessed in relation to latent class membership defined for latent variables defined for SES (income, education, country of birth, job type, and marital status), care barriers (insurance, urban/rural status, job loss, self-reported financial barriers to care, self-reported transportation barriers to care), and care use (pre-diagnostic regular care, breast cancer screening, mode of initial cancer detection (mammogram vs. noticed lump), and travel (based on estimated driving time) to diagnosis and surgery). Estimates are relative frequency differences (RFDs) and 95% confidence intervals, which compare frequency of delayed treatment for a given latent class with the indicated reference group. Results are presented (A) overall and (B) race-stratified; models are adjusted for age, stage, size, and grade at diagnosis.
(TIF)

**S3 Fig. Sensitivity analyses for prolonged treatment duration with stratification by ER and HER2 status ($N = 2,469$).** Prolonged treatment was assessed in relation to latent variables

defined for SES (income, education, country of birth, job type, and marital status), care barriers (insurance, urban/rural status, job loss, self-reported financial barriers to care, self-reported transportation barriers to care), and care use (pre-diagnostic regular care, breast cancer screening, mode of initial cancer detection (mammogram vs. noticed lump), and travel (based on estimated driving time) to diagnosis and surgery). Contrast estimates are relative frequency differences (RFDs) and 95% confidence intervals, which compare frequency of prolonged treatment for a given latent class with the indicated reference group. Results are stratified by (A) ER status and (B) HER2 status; models are adjusted for age, stage, size, and grade at diagnosis.
(TIF)

**S4 Fig. Linear-risk regression models for Oncotype Dx testing ($N$ = 1,597).** OncotypeDx was assessed among participants with ER+, HER2- disease in relation to latent variables defined for SES (income, education, country of birth, job type, and marital status), care barriers (insurance, urban/rural status, job loss, self-reported financial barriers to care, self-reported transportation barriers to care), and care use (pre-diagnostic regular care, breast cancer screening, mode of initial cancer detection (mammogram vs. noticed lump), and travel (based on estimated driving time) to diagnosis and surgery). Frequency and percentage of delayed diagnosis is reported in "n(%)" columns. Contrast estimates are relative frequency differences (RFDs) and 95% confidence intervals, which compare frequency of being untested for a given latent class with the indicated reference group. Results are presented (A) overall and (B) race-stratified; models are adjusted for age, tumor size, and grade at diagnosis.
(TIF)

## Author Contributions

**Conceptualization:** Matthew R. Dunn, Melissa A. Troester.

**Data curation:** Matthew R. Dunn.

**Formal analysis:** Matthew R. Dunn.

**Funding acquisition:** Melissa A. Troester.

**Methodology:** Matthew R. Dunn, Didong Li, Marc A. Emerson, Caroline A. Thompson, Hazel B. Nichols, Sarah C. Van Alsten, Mya L. Roberson, Stephanie B. Wheeler, Lisa A. Carey, Terry Hyslop, Jennifer Elston Lafata, Melissa A. Troester.

**Supervision:** Jennifer Elston Lafata, Melissa A. Troester.

**Visualization:** Matthew R. Dunn.

**Writing – original draft:** Matthew R. Dunn.

**Writing – review & editing:** Matthew R. Dunn, Didong Li, Marc A. Emerson, Caroline A. Thompson, Hazel B. Nichols, Sarah C. Van Alsten, Mya L. Roberson, Stephanie B. Wheeler, Lisa A. Carey, Terry Hyslop, Jennifer Elston Lafata, Melissa A. Troester.

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
