## [Editor Report · Decision Letter 0]

20 Jun 2024

Dear Dr Dunn, 

Thank you for submitting your manuscript entitled "The heterogenous impact of healthcare access factors throughout the breast cancer care continuum" for consideration by PLOS Medicine.

Your manuscript has now been evaluated by the PLOS Medicine editorial staff as well as by an academic editor with relevant expertise and I am writing to let you know that we would like to send your submission out for external peer review.

We also have the following editorial request: Please revise your title according to PLOS Medicine's style. Your title must be nondeclarative and not a question. It should begin with main concept if possible. "Effect of" should be used only if causality can be inferred, i.e., for an RCT. Please place the study design ("A randomized controlled trial," "A retrospective study," "A modelling study," etc.) in the subtitle (ie, after a colon). Please also add the location where your study took place (in your case United States) to the title.

Please re-submit your manuscript within two working days, i.e. by Jun 24 2024 11:59PM.

Kind regards,

Katrien G. Janin, PhD

Senior Editor

PLOS Medicine

---

## [Decision Letter · Decision Letter 1]

29 Aug 2024

Dear Dr Dunn,

Many thanks for submitting your manuscript "The impact of healthcare access factors throughout the breast cancer care continuum: an analysis of a racially diverse cohort of breast cancer survivors in North Carolina, USA" (PMEDICINE-D-24-01883R1) to PLOS Medicine. The paper has been reviewed by subject experts and a statistician; their comments are included below and can also be accessed here: [LINK]

After discussing the paper with the editorial team and an academic editor with relevant expertise, I'm pleased to invite you to revise the paper in response to the reviewers' comments. We plan to send the revised paper to some or all of the original reviewers, and we cannot provide any guarantees at this stage regarding publication.

We ask that you submit your revision by Sep 19 2024 11:59PM. However, if this deadline is not feasible, please contact me by email, and we can discuss a suitable alternative.

Don't hesitate to contact me directly with any questions (kjanin@plos.org). 

Best regards, 

Katrien 

Katrien Janin, PhD 

Associate Editor

PLOS Medicine

kjanin@plos.org

Comments from the academic editor:

Please also looked at the pattern between latent classes (or only generated one set of latent class that included SES, care barriers, and care use), which would provide a richer analyses and go a little bit deeper to bring out the mechanisms of disparities more clearly.

Comments from the reviewers: 

Reviewer #1: This manuscript describes the result of an analysis of the Carolina Breast Cancer Study cohort of approximately 3000 patients diagnosed 2008-2013. The investigators found racial differences in the care provided to patients with newly diagnosed breast cancer using four outcomes: having a later stage at diagnosis, having a longer time to treatment initiation, having a longer time to treatment completion, and being less likely to have OncotypeDx testing. The analysis also looked at the relationship between three potential latent class predictors (SES, barriers to care, and use of care) and these outcomes, finding that SES and, to a lesser extent care-barriers, were associated with inferior outcomes. Previous studies have demonstrated similar disparities with breast cancer care. This analysis provides interesting findings relative to several clinically relevant outcomes using latent class variables. In fact, the paper is as much about deriving the latent class variables as it is about describing the associations between those variables and the selected outcomes. Several concerns regarding the methods and discussion were noted, as detailed below. 

An 8-month threshold was used for the treatment duration outcome among chemotherapy recipients. How did the analysis handle patients with HER2-positive and/or stage IV breast cancer? Patients with HER2 positive breast cancer typically receive trastuzumab based chemotherapy, where the recommended duration of treatment is at least 12 months. And patients with stage IV breast cancer could receive chemotherapy for >8 months if it is controlling the cancer and not causing bothersome side effects. In either case, treatment would extend beyond the 8-month threshold and could mean that all patients in these groups experienced delayed treatment completion. 

The proportion of patients with at least one delay is high - about three quarters. Considering this, it could be interesting to conduct a sensitivity analysis using longer thresholds for delay. Also, it is possible that some outcomes could have been highly correlated with each other. For example, the same patients could have experienced a delayed diagnosis and a prolonged treatment (the predictors for these outcomes were similar). Did the analysis look for correlation between the outcomes?

The approach focuses on a threshold analysis - treatment initiation within 30 days, treatment completion within 2-8 months, etc. This is relatively easy to interpret but makes it hard to understand how much risk is associated with the delay. For example, a two-day delay would have different implications than a 62-day delay. Did the team consider also describing the difference in the median time-to-event?

The analysis defines four outcomes: delayed diagnosis, delayed treatment initiation, delayed treatment completion, and use of oncotypedx. In the text and in Figure 1, the analysis aggregates the first three outcomes into a composite 'any delay' outcome. Consider defining this aggregate outcome in the methods section. Was there a difference in the number of potential delay-events/patient for black vs. non-black patients? Also, compared to the other outcomes assessed by this study, OncotypeDx seems like an outlier. Why should OncotypeDx be included, especially considering that another analysis (reference 61) assessed a similar outcome using the CBCS database. 

Other potential limitations include that the study only included patients from North Carolina and that all patients were diagnosed 10+ years ago. 

The title, "The heterogeneous impact of healthcare access factors throughout the breast cancer care continuum," is confusing and potentially misleading. The analysis includes four outcomes, all focusing on the initial diagnosis/treatment phase of care. No outcomes address other aspects of the care continuum (e.g., survivorship, end of life). If 'throughout the care continuum' is intended to refer to all stages of breast cancer, a different description could be more accurate. The title describes a heterogeneous impact, but it is not altogether clear what is meant by heterogeneity of effect or that this is the major finding of the paper. The title focuses on healthcare access whereas the paper includes a broad set of potential predictors (i.e., SES, care access, care use) which may or may not impact the accessibility of the healthcare system. And the title does not touch on racial disparities, which is a large focus of the analysis and manuscript. 

Reviewer #2: 

1. This study included the first primary breast cancer diagnosis between 2008 and 2013 from CBCS 3. Do the authors have the background data on the possibility of women with breast cancer who were never diagnosed and subsequently not included in this cohort (the original pool from which samples in this manuscript are drawn)? I imagine this neglected group may be from a very different background than those included in the cohort. 

2. For the sentence "Given that results did not change when restricting to stage I-III cancer (N=2,886), we elected to include patients of all stages": this statement shows an outcome-oriented approach, I would suggest re-phrasing to using a primary analysis set including all stage patients, plus a sensitivity analysis set including only stage I-III cancer and proving the rational for this sensitivity analysis. 

3. Moreover, have the authors considered a sensitivity analysis comparing black patients with non-Hispanic White patients (the majority of the non-black patient group)? 

4. SES information was obtained approximately 5 months after their breast cancer diagnosis, can the authors give more details on the time interval from diagnosis to SES information collection? The reason for this is that diagnosis and treatment of breast cancer may potentially change their SES. 

5. It would be very helpful to understand who the patients are by providing the demographic information, including age, tumor size, stage, and grade at diagnosis, by race group.

6. Table 1: Can the authors provide the breakdown variables of the SES, care barrier and care use factors by race group? If limited by word/table count, this can be shown in the supplementary document. 

7. Figure 1 is very informative and gives an overall picture of the delays. Can the authors add the % of delays (diagnosis stage, initiation delay and prolonged duration) into the figure? 

8. Supplementary Table 1 shows the outcome distribution by group, it is the very first description and modelling results of the study outcome variables, and it should be included in the main manuscript. 

9. Figure 2-5: Could the authors add the count and raw percentage of the delays in each SES/Care barrier/use category? It provides the details of the delays in each subgroup, and if a complete data approach is used in the linear-risk regression, the results exclude patients with missing values and thus use fewer patients. 

10. It seems that by design, the causes are closely correlated with each other, e.g., judging by common sense, one with low SES is probably more likely to have more care barriers and low care use, same for the outcome variables, when one's diagnosis is neglected for some reasons, the same reasons may also play a role in the subsequent treatment. The authors performed their analysis using individual models for one cause on one outcome, separately. Can the authors further indicate how the causes (SES, care barriers and care use) are correlated with each other, and how the outcomes (delay in diagnosis, treatment initiation and prolonged treatment) are correlated with each other? It would be interesting and important to see if the overall effect of a compound variable/cause integrating SES, care barriers and care use together.

Reviewer #3: The article "The heterogenous impact of healthcare access factors throughout the breast cancer care continuum" presents associations between latent measures of access to healthcare with timeliness of diagnosis, initiation and completion of treatment; between access to healthcare and receipt of the Oncotype DX genomic test; and describes the differences in the timeliness of care between black and non-black patients.

-

The study presents relevant findings for the academic community and health systems, corroborating findings already described elsewhere in the world, and which bring to the table an important discussion related to disparities in access to health services between black and non-black women with breast cancer.

-

Below are considerations about the reviews: the article mentions some variables consolidated and studied in another published study of the Carolina Breast Cancer Study, but it is not clear how these variables could be part of this study without being properly described; 

It does not describe the organization of the North Carolina health system, which would be important to understand, for example, how these barriers are configured and whether they are related to the health system, public policies or the living conditions of these women. What should the North Carolina health system guarantee and does not guarantee? since the health systems may consider these findings informative in designing algorithms for

identifying and triaging patients for resource-limited patient navigation programs.

-

Please include the period of data collection and explain how the data was organized? Where it was exported from?

-

There are differences in the income values defined to consider the socioeconomic level of women in the outcome. It was not clear why different values were used.

"SES factors. SES information was obtained from an in-home survey administered to participants

approximately 5 months after their breast cancer diagnosis. Each variable was defined as a binary or

multinomial categorical variable: household income (<$15,000, $15,000-50,000, or $50,000+"

"Therefore, we evaluated each care outcome in relation to household income (<20k, 20-50k, 50k+)"

-

Figure 1 needs to use the terms homogeneously to prevent interpretation by the reader: "initiation" is "treatment initiation" "completion" or "duration".

---

* Please upload any figures associated with your paper as individual TIF or EPS files with 300dpi resolution at resubmission; please read our figure guidelines for more information on our requirements: http://journals.plos.org/plosmedicine/s/figures. While revising your submission, please upload your figure files to the PACE digital diagnostic tool, https://pacev2.apexcovantage.com/. PACE helps ensure that figures meet PLOS requirements. To use PACE, you must first register as a user. Then, login and navigate to the UPLOAD tab, where you will find detailed instructions on how to use the tool. If you encounter any issues or have any questions when using PACE, please email us at PLOSMedicine@plos.org.

OBSERVATIONAL STUDIES

* Abstract: Please include the study design, population and setting, number of participants, years during which the study took place (enrollment and follow up), length of follow up, and main outcome measures.

* Please ensure that the study is reported according to the STROBE (or appropriate STOBE extension) guideline (available from: https://www.equator-network.org/reporting-guidelines/strobe) and include the completed STROBE (or STROBE extension) checklist as Supporting Information. Please add the following statement, or similar, to the Methods: "This study is reported as per the Strengthening the Reporting of Observational Studies in Epidemiology (STROBE) guideline (S1 Checklist)." When completing the checklist, please use section and paragraph numbers, rather than page numbers. 

FIGURES AND TABLES

SUPPLEMENTARY MATERIAL

REFERENCES

---

## [Decision Letter · Decision Letter 2]

29 Oct 2024

Dear Dr. Dunn,

Thank you very much for re-submitting your manuscript "A latent class assessment of healthcare access factors and disparities in breast cancer care timeliness" (PMEDICINE-D-24-01883R2) for review by PLOS Medicine.

Thank you for your detailed response to the editors' and reviewers' comments. I have discussed the paper with my colleagues and the academic editor, and it has also been seen again by the statistical reviewer. The changes made to the paper were mostly satisfactory to the reviewer. As such, we intend to accept the paper for publication, pending your attention to the reviewer's and editors' comments below in a further revision. When submitting your revised paper, please once again include a detailed point-by-point response to the editorial comments.

[LINK]

In revising the manuscript for further consideration here, please ensure you address the specific points made by each reviewer and the editors. In your rebuttal letter you should indicate your response to the reviewers' and editors' comments and the changes you have made in the manuscript. Please submit a clean version of the paper as the main article file. A version with changes marked must also be uploaded as a marked up manuscript file. Please also check the guidelines for revised papers at http://journals.plos.org/plosmedicine/s/revising-your-manuscript for any that apply to your paper.

We ask that you submit your revision within 1 week (Nov 05 2024). However, if this deadline is not feasible, please contact me by email, and we can discuss a suitable alternative.

Please do not hesitate to contact me directly with any questions (atosun@plos.org). If you reply directly to this message, please be sure to 'Reply All' so your message comes directly to my inbox.

We look forward to receiving the revised manuscript.

Sincerely,

Alexandra Tosun, PhD

Associate Editor 

PLOS Medicine

plosmedicine.org

Comments from Reviewers:

Reviewer #2: Line 249, the newly added content "Effect estimates were similar in sensitivity analyses with Black 249 and White patients only (N=2,916) (S1 Table)" should be moved to the Results section. 

Line 272, the newly added content "Sensitivity analyses showed that results were consistent when restricting to stage I-III patients (S3 Table)" should be moved to the Results section.

The manuscript in its current version uses both Roman numbers and Arabic numbers for cancer stages, can the authors use one form consistently? 

For Figures 2-4, It is great that the authors added the N(%), it is important to have them there, because 1). It is often good to have the Ns to understand the distribution; 2) without the N (%), the RFD by subgroup would be very misleading, as the reference groups are different between the two subgroups. Can the authors adjust the title on the right "Non-black" and "Black" so that they are above the columns of both N(%) and RFD? 

Figure 5 is an interesting one, the delays (diagnosis/treatment/treatment) do not show any strong pattern across the latent class groups (SES, carer barriers, travel), however, the distribution across the latent class groups themselves shows a strong pattern, that is, only very few people in high SES had barriers (all the grey patches). This might be what Reviewer 1 indicated in the comment "…this paper is as much about deriving the latent class variables as it is about describing the associations between those variables and the selected outcomes." I agree with this.

[LINK]

Requests from Editors:

CODE AVAILABILITY

Thank you for agreeing to depositing code in Github prior to publication. Please add a sentence to your data availability statement (in the online submission form) regarding the code used in the study, e.g. "The code used in the analysis is available from Github [URL]”. Please note that because Github depositions can be readily changed or deleted, we encourage you to make a permanent DOI'd copy (e.g. in Zenodo) and provide the URL. 

FINANCIAL DISCLOSURE 

Please ensure to update the financial disclosure section in the online submission form with the details provided in lines 374-376. 

ABSTRACT

1) l.91: Please define ‘ER+’ and ‘HER2-‘ at first use, for example: “evaluated among patients with early stage, ER+ (estrogen receptor), HER2- (human epidermal growth factor receptor 2) disease”. 

2) l.94: CBCS or CBCS3? Please clarify.

3) l.101ff: Throughout, we suggest reporting statistical information as follows to improve clarity for the reader "5.5% (95% CI [2.4,8.5])". When reporting 95% CIs please separate upper and lower bounds with commas instead of hyphens as the latter can be confused with reporting of negative values. Please revise throughout the entire manuscript.

4) In the last sentence of the Abstract Methods and Findings section, please describe the main limitation(s) of the study's methodology.

5) Please include the important dependent variables that are adjusted for in the analyses, similar to what is reported on lines 312-315.

6) We suggest explaining in the Abstract that "travel" was measured as a function of time.

AUTHOR SUMMARY

1) l.119: Please define ‘SES’ at first use.

2) In the final bullet point of 'What Do These Findings Mean?', please include the main limitations of the study in non-technical language.

INTRODUCTION

1) l.150: Please define ‘U.S.’ at first use.

2) l.161, please change to “Black women and women with low-SES”. Please note that PLOS Medicine prefers the use of patient-centered language.

METHODS AND RESULTS

1) l.182: Please define ’USA’ at first use and ensure to use a consistent format (earlier you used U.S.). Please revise throughout.

2) l.191: Please provide the approval number if one was obtained.

3) l.217: When reporting age, please add a unit, such as ‘years’ (age 40 years). Please revise throughout.

4) ll.239-240: “Effect estimates were similar in sensitivity analyses with Black and White patients only (N=2,916) (S1 Table).” – Do you mean clinical characteristics were similar between Black and White patients? The n-number also does not match the number in S1 Table (n=2998). Please check and revise.

5) l.241: Please define ‘AJCC’.

6) l.244: You state that “progesterone receptor (PR) status” was obtained from pathology reports, but it is not reported within the manuscript. Please clarify. 

7) l.257: Please define ‘HR’ at first use.

8) l.260: “stage IV” – please see the reviewer comment regarding Arabic and Roman numerals and revise accordingly. 

9) l.266, please change to: “that time greater than 38 weeks”

10) l.276: Please note that you alternate between using ‘n’ and ‘N’ when reporting total numbers. Please revise using a consistent format.

11) ll.330-331, we suggest specifying as follows: “Clinical and demographic characteristics, including details on care barriers and care use, of the study population are reported in S1 and S2 Tables.”

12) l.332: For better comprehension, we suggest repeating the total n-number here. For example: “Overall out of 2,998 CBCS participants, 18% had a delayed diagnosis,…”

13) Table 1: We suggest adding the total n-number for each group (Overall, non-Black, Black) at the top of the table. Also, please add a definition for ‘travel’ below the table, i.e. time versus distance, what Oncotype DX is, and define high, moderate and low SES briefly below the table.

14) Figure 1: Please add the total n-number to each group in the figure.

15) ll.368-369: “Black women more frequently were classified as having more barriers (24% v 10%; RFD= 13.8%, CI: 11.2,16.5). – compared to? Please clarify.

16) Figure 2/3/4/5: Please add a definition for ‘travel’ below the table, i.e. time versus distance, and briefly define high, moderate and low SES as well as what ‘barriers’ entails.

17) ll.428-430: “There was also greater frequency of prolonged treatment among participants with more care barriers (54% vs 47%; RFDadj = 7.3%, CI: 2.4, 12.2).” - compared to? Please clarify and revise throughout for clarity.

18) ll.449: When reporting percentages, as here "46% delayed diagnosis among moderate SES", please be sure to provide a denominator to help guide the reader. Please revise throughout and add as appropriate.

19) Figure 5: Please consider avoiding the use of red (particularly in contrast to grey) in order to make your figure more accessible to those with color blindness.

20) ll.464-466: Please revise for clarity.

21) Table 2: Please define ‘CI’. Please add a unit for income.

22) When revising the figures and tables, including supplementary tables and figures, please keep in mind that figures and tables should be self-explanatory on a stand-alone basis, i.e. sufficient detail is required in the descriptions/legends. 

DISCUSSION

Please remove the ‘Conclusion’ subheading from the Discussion.

REFERENCES

Where website addresses are cited, please use the word ‘accessed’ when specifying the date of access (e.g. [accessed: 12/06/2024]).

SUPPLEMENTARY MATERIAL

In the published article, supporting information files are accessed only through a hyperlink attached to the captions. For this reason, you must list captions at the end of your manuscript file. You may include a caption within the supporting information file itself, as long as that caption is also provided in the manuscript file. Do not submit a separate caption file.

When SI files are contained with a single file:

Please label the file as ‘S1 Supporting Information’.

Please apply alphabetical labelling to each table and figure contained within the S1 file. For example, ‘Fig A’ to ‘Fig Z’ and ‘Table A’ to ‘Table Z’.

Plain text does not need to be labelled and can just be given a title as necessary. For example, ‘Statistical Analysis Plan’.

Please cite tables/figures as ‘Fig A in S1 Supporting Information’ and/or ‘Table A in S1 Supporting Information’, for example.

Please cite plain text as, ‘Statistical Analysis Plan in S1 Supporting Information’, for example.

When SI files are uploaded as separate files:

Please label tables as ‘S1 Table’ (so on) and figures as ‘S1 Fig’ (and so on).

Any additional documents (protocols/analysis plans etc.) can be labelled as ‘S1 Protocol’, for example. Please cite items as exactly as labelled.

SOCIAL MEDIA

To help us extend the reach of your research, please provide any X (formerly known as Twitter) handle(s) that would be appropriate to tag, including your own, your co-authors’, your institution, funder, or lab. Please enter in the submission form any handles you wish to be included when we post about this paper.

General Editorial Requests

---

## [Editor Report · Decision Letter 3]

14 Nov 2024

Dear Dr Dunn, 

On behalf of my colleagues and the Academic Editor, Aaloke Mody, I am pleased to inform you that we have agreed to publish your manuscript "A latent class assessment of healthcare access factors and disparities in breast cancer care timeliness" (PMEDICINE-D-24-01883R3) in PLOS Medicine.

I appreciate your thorough responses to the reviewers' and editors' comments throughout the editorial process. We look forward to publishing your manuscript, and editorially there are only a few remaining minor stylistic/presentation points that should be addressed prior to publication. We will carefully check whether the changes have been made. If you have any questions or concerns regarding these final requests, please feel free to contact me at atosun@plos.org.

Please see below the minor points that we request you respond to:

1) Figure 5: Please define ‘Var’ or spell out.

2) Figure 2/3/4/5: In the figure description, please define ‘SES’ (at first use).

Before your manuscript can be formally accepted you will need to complete some formatting changes, which you will receive in a follow up email (including the editorial point above). Please be aware that it may take several days for you to receive this email; during this time no action is required by you. Once you have received these formatting requests, please note that your manuscript will not be scheduled for publication until you have made the required changes.

PRESS

Sincerely, 

Alexandra Tosun, PhD 

Associate Editor 

PLOS Medicine